# Occupational Diseases among Workers in Lower and Higher Socioeconomic Positions

**DOI:** 10.3390/ijerph15122849

**Published:** 2018-12-13

**Authors:** Henk F. van der Molen, Sanne de Vries, Judith K. Sluiter

**Affiliations:** Amsterdam UMC, Coronel Institute of Occupational Health, Netherlands Center for Occupational Diseases, Amsterdam Public Health research institute, PO Box 22660, 1100 DD Amsterdam, The Netherlands; s.de.vries@scp.nl (S.d.V.); coronel@amc.nl (J.K.S.)

**Keywords:** occupational disease, socioeconomic position, incapacity for work

## Abstract

*Background:* To determine differences between workers in lower and higher socioeconomic positions (SEP) in incidences of occupational disease (OD) and incapacity for work due to ODs. *Methods:* From a Dutch dynamic prospective cohort of occupational physicians (OPs), ODs assessed by OPs were retrieved for lower and higher SEP groups. *Results:* Among the lower SEP, musculoskeletal disorders, and noise-induced hearing loss (NIHL) comprised two-thirds of the OD diagnoses. Among the higher SEP, stress/burnout comprised 60% of the OD diagnoses. Temporary and permanent incapacity for work due to work-related lower back disorders and repetitive strain injuries differed significantly between workers in lower compared to higher SEP. *Conclusions:* Occupational diseases occur at a 2.7 higher incidence rate for workers in lower SEP compared with higher SEP. Incapacity for work varies between the type of OD and the level of SEP.

## 1. Introduction

Maintaining the health of workers in lower socioeconomic positions (SEP) is an increasing cause for concern [1,2]. In higher SEP jobs, however, task demands can also exceed individual capacities, as shown in a recent European overview regarding the effect of work on mental health [3]. Due to demographic changes, the age of workers is increasing, resulting in more frequent morbidity or, comorbidity, sickness absence [4], lower physical capacities, and higher cumulative work exposures. These changes increase the risk of work-related diseases [5,6].

Socioeconomic status as a determinant of health has been recognized widely and analysed as a composition of various underlying aspects such as education, income and social position [7]. Occupation is one dimension of socioeconomic status, and the weighing of occupations on education and income by job title [8] could reflect SEP. Job complexity could also be an indicator of SEP, with managerial and intellectually complex jobs at the higher end of SEP, and manual, less complex jobs at the lower end of SEP [9]. Ultimately, lower and higher SEP will reflect the distinction between blue and white-collar workers, with different task demands and working circumstances. 

Registries of occupational diseases (ODs) can provide insight into job title, work-related diseases and incapacity for work. ODs are relevant outcomes, because they are clinically established diseases mainly caused by work [10,11]. In the Netherlands, OPs are obliged to report ODs to the Netherlands Center for Occupational Diseases (NCOD). In contrast to most other countries, there is no financial compensation system for diagnosed ODs in the Netherlands [12]. An OD is defined as a disease for which the work-related fraction is >50%. Each worker diagnosed with an OD is anonymously reported to the NCOD, with the following information recorded in its database: disease or pathology with clinical diagnosis, demographic characteristics, exposure, job title, economic sector, and incapacity for work [12].

It is hypothesized that workers in lower and higher SEP still differ in health disparities from an occupational perspective, with a higher burden of work-related diseases for lower SEP. The objective of this study is to determine differences between workers in lower and higher SEP occupational groups in (i) incidence and type of occupational disease and (ii) incapacity for work as a consequence of an occupational disease.

## 2. Methods

From a dynamic prospective cohort of occupational physicians (OPs), all ODs diagnosed by the OPs and reported to the Netherlands Centre for Occupational Diseases (NCOD) over a seven-year period (2010–2016) were retrieved for workers with lower and higher SEP. Elementary occupations and machine operating and assembly jobs (ISCO-08 groups 81–83, 91–96) were defined as lower SEP. Managerial and professional intellectual jobs (ISCO-08 groups 11–14, 21–26) were defined as higher SEP. The population size for the higher and lower SEP was retrieved from the database of Eurostat [13], consisting of 1,090,700 workers for the lower SEP and 2,642,00 workers for the higher SEP in 2016. During the period of 2010–2016, the annual populations varied between 1,002,400 and 1,207,000 workers for the lower SEP and between 2,478,800 and 2,642,000 workers for the higher SEP group.

An OD was defined as a clinically diagnosed disease (ICD-10 classification) that was predominantly caused by work-related factors [12]. Incidences were determined for all ODs and seven frequently occurring groups of ICD diagnosis, which were the following: noise-induced hearing loss (NIHL): H833, H919; non-specific low back pain: M545; repetitive strain injuries of the upper extremity: G560, G589, I730, M189, M199 (partly), M531, M700, M709 (partly), M770, M771; arthrosis of the knee and hip: M169, and M179; burnout/stress: Z730, F432; contact dermatitis: L239, L249, and L259; asthma/COPD: J439, J449, J450, J451, J458, J459, and J689.

Incidence of OD with incapacity for work was determined for the seven frequently occurring ODs. Trends in incidence were estimated using a multilevel negative binomial regression model. Annual incidence was determined by dividing the number of reported ODs in 2016 by the total working population. Differences in incapacity for work between lower and higher SEP were determined through chi-squared tests (IBM SPSS 24, IBM, Armonk, NY, USA). Trends in incidence rates were analysed with multilevel binomial regression analysis [14]. Case counts were analysed using a negative binomial regression model with 2010 as the reference year. Population estimates, as natural logarithms of the annual number of workers, were included in the regression model as an “offset”. No report of a specific OD diagnosis was assumed to indicate a report of zero cases for that year. Trend analyses were performed with Stata 15.1 (StataCorp LLC, Texas, TX, USA). 

## 3. Results 

In total, 1233 OPs reported 13,917 ODs for workers in lower and higher SEP during the period of 2010–2016 (see Table 1). Lower SEP consisted of 8145 workers (82% male), of which 82% were aged over 40 years. Higher SEP consisted of 5772 workers (59% male), of which 71% were aged over 40 years. 

Among the lower SEP, musculoskeletal disorders (37%) and NIHL (32%) comprised two-thirds of the OD diagnoses, with decreasing trends for non-specific low back pain (−12%; 95% CI: −18% to −6%) and NIHL (−7%; 95% CI: −11% to −3%). Among the higher SEP, stress/burnout comprised 60% of the OD diagnoses, with an increasing trend (6%; 95% CI: 3%−8%).

Incapacity for work varied between OD and SEP. It was rarely reported for NIHL (2%−3%), with no significant differences (*p* = 0.177) between SEP. Highest (temporary) incapacity for work was reported for stress/burnout (94%), with no significant differences (*p* = 0.382) between SEP. Occupational contact dermatitis, COPD and asthma, and knee and hip arthrosis showed the highest risks for permanent incapacity for work, varying from 13% to 32%, but no significant differences between SEP (*p* > 0.50). Incapacity for work due to work-related lower back disorders (69% vs. 9%) and repetitive strain injuries (89% vs. 47%) differed significantly (*p* = 0.000) between lower and higher SEP.

In 2016, 54 per 100,000 workers in Dutch elementary occupations, machine operating and assembly jobs, and managerial and professional jobs had an OD diagnosed, of which 98 per 100,000 workers were from lower SEP and 36 per 100,000 workers were from higher SEP.

## 4. Discussion

Occupational diseases occurred at a 2.7 higher incidence rate for workers in lower SEP compared with higher SEP in 2016. Among the workers in a lower SEP, musculoskeletal disorders and noise-induced hearing loss were the most frequently occurring ODs, with trends decreasing over a seven-year period. Among the workers in a higher SEP, stress/burnout was the most frequently occurring OD, with an increasing trend over a seven-year period. The highest temporary incapacity for work was reported for work-related stress/burnout, with no differences between SEP. Work-related knee and hip arthrosis, as well as dermal and lung diseases showed the highest risks for permanent incapacity for work, both in lower and higher SEP. Temporary and permanent incapacity for work due to work-related musculoskeletal disorders were higher in lower SEP workers compared with higher SEP workers.

Misclassification, selection bias, and uncertainty about missing values are methodological limitations in this study. Misclassification in SEP due to job title by ISCO-08 codes is possible, but considered to be non-differential. Selection bias may have been introduced due to differences in entrance to OPs; however, most employed workers visit their OPs when incapacity for work occurs or as a result of worker health monitoring, irrespective of SEP. Furthermore, the reporting of ODs may differ between SEP, e.g., OPs are more familiar with work-related musculoskeletal disorders, which may cause selection bias. Since many ODs emerge after cumulative exposures over time, the incidence of OD may depend on age; however, in our study no large differences in age between lower and higher SEP were found at the time of the assessed ODs. Uncertainty about missing values in regression analyses may have a large impact on the estimates of trends in OD incidence. To counteract this, we modified our methods to input zero reports only when an OP demonstrated being active in a specific year, thus assuming true zero reports [15]. The data in this paper were collected from the Dutch national registry. In future research, OD incidence and incapacity for work between lower and higher SEP could be enriched if SEP at an individual level also encompasses educational and income level. 

Occupational diseases occur in both higher and lower SEP, although they differ in incidence, type of OD, and consequences for work capacity. Our results suggest that among lower SEP, hazardous workplaces and adverse work practices are still present, with known biomechanical, physical, and chemical risks factors still to be eliminated. In higher SEP requiring managerial and intellectual tasks, psychosocial demands seem to present the largest risk factors for mental disorders, accompanied by a temporary incapacity for work. Temporary and permanent incapacity for work due to work-related musculoskeletal disorders are higher for lower SEP compared with higher SEP, suggesting fewer opportunities to modify work tasks and working circumstances for lower SEP jobs. These findings are in line with evidence that work indeed explains socioeconomic inequalities in self-rated health among workers besides lifestyle factors [16], while differences in incidences of the specific ODs reflect the prevalence of risk factors in the lower SEP (e.g., noise levels [17] and biomechanical risk factors [18] and higher SEP (psychosocial risk factors [19]). 

The understanding of whether SEP influences the possibility to stay in work with an OD is of considerable interest. Although adverse health conditions due to work hamper sustainable work ability, especially in lower SEP [20], higher SEP is at risk for especially work-related stress disorders. The frequently reported ODs among the Dutch working population revealed population attributable fractions varying between 3% and 25%, leaving considerable potential for taking preventive actions on work-related exposure in terms of the occurrence of ODs and their medical and productivity cost [21].

## 5. Conclusions

Occupational diseases occur at a 2.7 higher incidence rate for workers in lower SEP compared with higher SEP. Occupational diseases differ in incidence, type, and resulting incapacity for work between workers in higher and lower SEP. Among the workers in a lower SEP, musculoskeletal disorders and noise-induced hearing loss were the most frequently occurring ODs, with trends decreasing over a seven-year period. Among the workers in a higher SEP, stress/burnout was the most frequently occurring OD, with an increasing trend over a seven-year period. This provides further insight into disparities between workers in different socioeconomic positions in terms of their capacity to counteract negative consequences due to work. 

## Figures and Tables

**Table 1 ijerph-15-02849-t001:** Occupational diseases and incapacity for work among workers in lower and higher socioeconomic positions (SEP).

International Classification of Diseases—10	N 2010–2016	Incidence per 100,000 2016	2010–2016		Incapacity for Work 2010–2016	
	Occupational Diseases		IRR *	95% CI	Temporary%	Permanent %
**Total**						
Lower SEP	8145	98	0.95	0.94–0.97		
Higher SEP	5772	36	1.03	1.01–1.05		
**Noise Induced Hearing Loss**						
Lower SEP	2599	23	**0.93**	**0.89–0.97**	0.8%	2%
Higher SEP	830	3	0.99	0.93–1.05	0.2%	2%
**Non-specific Low Back Pain**						
Lower SEP	635	6	**0.88**	**0.82–0.94**	**64%**	**5% **
Higher SEP	131	0.5	0.92	0.80–1.04	**7%**	**2%**
**Arthrosis Knee/Hip**						
Lower SEP	157	2	0.97	0.88–1.07	40%	27%
Higher SEP	19	0.1	0.97	0.71–1.22	32%	21%
**Repetitive Strain Injury**						
Lower SEP	2192	31	1.00	0.97–1.03	**83%**	**6%**
Higher SEP	339	2	0.95	0.89–1.02	**45%**	**2%**
**Stress/Burnout**						
Lower SEP	463	8	1.00	0.95–1.06	91%	3%
Higher SEP	3439	25	**1.06**	**1.03–1.08**	92%	2%
**Contact Dermatitis**						
Lower SEP	193	2	1.01	0.94–1.09	46%	13%
Higher SEP	28	0.1	1.09	0.88–1.29	36%	18%
**COPD/Asthma**						
Lower SEP	54	0.5	0.93	0.79–1.07	44%	32%
Higher SEP	20	-	0.84	0.60–1.07	40%	30%

* IRR = Incidence Rate Ratio (reference year 2010). Bold means *p* < 0.05.

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
