# Peer review of "Occupational Diseases among Workers in Lower and Higher Socioeconomic Positions"

_ijerph, 2018, doi:10.3390/ijerph15122849_

Round 1

Reviewer 1 Report

The aim of the study is very interesting, reporting about an important field of public health such as the health of workers in lower socioeconomic positions. Each part of the manuscript is good. Therefore, I have no remarks to make.

Author Response

Response to reviewer 1

The aim of the study is very interesting, reporting about an important field of public health such as the health of workers in lower socioeconomic positions. Each part of the manuscript is good. Therefore, I have no remarks to make.

Response 1: Thank you for your compliment and time taken to review our manuscript.

Reviewer 2 Report

This study investigated the discrepancy on the incidence of occupational diseases between workers in lower socioeconomic position and workers in higher socioeconomic position. A dynamic prospective cohort study was used in which all occupational diseases were diagnosed by the occupational physicians and reported to the Netherlands Centre for Occupational Diseases. The results showed that the workers in lower socioeconomic position had a higher incidence rate of occupational diseases than the workers in higher socioeconomic position. It was also found that the workers in higher socioeconomic position had a higher incidence of mental illnesses than the workers in lower socioeconomic position, and the workers in lower socioeconomic position had a higher incidence of physical illnesses than the workers in lower socioeconomic position. The content is useful to understand the incidence of various types of occupational diseases for worker in  higher and lower socioeconomic position, which provides a reference for the government and organizations to raise the awareness on the occupational safety and health of worker. The study was well designed and executed and all these findings are interesting.

However, more detailed explanations and references can be added to support the findings. This paper can be accepted after the authors have addressed the comments and revised accordingly.

Some specific comments are given below.

Main queries

In page 1 (line 29-30), the      authors stated the definition of socioeconomic status, “Socioeconomic      status as a determinant of health … education, income and social      position”. In the same page (line 32-35), the authors defined higher SEP      and lower SEP, “Job      complexity could also be an indicator of SEP, with managerial and      intellectually complex jobs at the higher end of SEP and manual, less      complex jobs at the lower end of SEP. Ultimately, lower and higher SEP      will reflect the distinction between blue and white-collar workers, with      different task demands and working circumstances.”

Actually, education and income should be the fundamental factors of SEP. However, the authors defined the higher SEP and lower SEP based on the features of the jobs and the job positions. There’s a confusion if using the wording, socioeconomic positions (SEP),  to describe the workers is appropriate or not. Whether such classification should be white-collar workers and blue-collar workers, or there are other better terms?

Why the      authors just selected these seven types of occupational diseases? Did it      imply that other occupational diseases were not significant to be      considered? The selection criteria should be well given.

Based      on the seven types of occupational diseases, the authors concluded that      lower SEP had a higher incidence of occupational diseases than higher SEP.      Such conclusion may not be persuasive enough. “Occupational diseases”      should not be used as the focus because occupational diseases should      include various types of illnesses and diseases.

In addition, some small mistakes and issues should be noticed.

In      Discussion, the authors can add some more explanations on why higher SEP      had a higher incidence of mental health problem than lower SEP, why lower      SEP had a higher incidence of physical illnesses than higher SEP, and why      distress/ burnout had a high percentage on temporary incapacity for work,      and asthma and arthrosis knee/ hip had high percentages on permanent      incapacity for work. All these need further elaborated and indepth      discussions.

In      Discussion (page 4, line 117-123), more references can be added to support      the findings and results.

Two      sections, Limitation and Practical implication, can be added to the paper      to enrich the contents.

The      conclusion (page 4) seems a bit short. A more detailed summary should be      made.

Author Response

Repsonse to reviewer 2

This study investigated the discrepancy on the incidence of occupational diseases between workers in lower socioeconomic position and workers in higher socioeconomic position. A dynamic prospective cohort study was used in which all occupational diseases were diagnosed by the occupational physicians and reported to the Netherlands Centre for Occupational Diseases. The results showed that the workers in lower socioeconomic position had a higher incidence rate of occupational diseases than the workers in higher socioeconomic position. It was also found that the workers in higher socioeconomic position had a higher incidence of mental illnesses than the workers in lower socioeconomic position, and the workers in lower socioeconomic position had a higher incidence of physical illnesses than the workers in lower socioeconomic position. The content is useful to understand the incidence of various types of occupational diseases for worker in  higher and lower socioeconomic position, which provides a reference for the government and organizations to raise the awareness on the occupational safety and health of worker. The study was well designed and executed and all these findings are interesting.

However, more detailed explanations and references can be added to support the findings. This paper can be accepted after the authors have addressed the comments and revised accordingly.

Response 1: Thank you for your compliment and time taken to review our manuscript.

Some specific comments are given below.

Main queries

In page 1 (line 29-30), the      authors stated the definition of socioeconomic status, “Socioeconomic status as a determinant of health … education, income and social position”. In the same page (line 32-35), the authors defined higher SEP and lower SEP, “Job  complexity could also be an indicator of SEP, with managerial and intellectually complex jobs at the higher end of SEP and manual, less      complex jobs at the lower end of SEP. Ultimately, lower and higher SEP will reflect the distinction between blue and white-collar workers, with      different task demands and working circumstances.” Actually, education and income should be the fundamental factors of SEP. However, the authors defined the higher SEP and lower SEP based on the features of the jobs and the job positions. There’s a confusion if using the wording, socioeconomic positions (SEP),  to describe the workers is appropriate or not. Whether such classification should be white-collar workers and blue-collar workers, or there are other better terms?

Response 2: We agree that the concept of socioeconomic status is multifactorial. However we could only choose one domain, i.e. job title. We elaborated in the discussion on this limitation:

‘The data in this paper are collected from a Dutch national registry. In future research OD incidence and incapacity for work between lower and higher SEP could be enriched if SEP at individual level encompass also educational and income level.

We prefer to remain the wording of socioeconomic position, however we acknowledged this limitation in the discussion paragraph.

Why the authors just selected these seven types of occupational diseases? Did it imply that other occupational diseases were not significant to be considered? The selection criteria should be well given.

Response 3: Incidences were determined for all ODs. But also for the seven most frequently occurring groups of ICD diagnosis, which were: Noise-Induced Hearing Loss (NIHL): H833, H919; Non-specific low back pain: M545; Repetitive strain injuries upper extremity: G560, G589, I730, M189, M199 (partly), M531, M700, M709 (partly), M770, M771; Arthrosis knee and hip: M169, M179; Burnout, stress: Z730, F432; Contact dermatitis: L239, L249, L259; Asthma/COPD: J439, J449, J450, J451, J458, J459, J689.

Based on the seven types of occupational diseases, the authors concluded that lower SEP had a higher incidence of occupational diseases than higher SEP. Such conclusion may not be persuasive enough. “Occupational diseases” should not be used as the focus because occupational diseases should include various types of illnesses and diseases.

Response 4: We prefer this wording, because the conclusion on the higher OD incidence in lower SEP is based on all ODs.

In addition, some small mistakes and issues should be noticed. In  Discussion, the authors can add some more explanations on why higher SEP had a higher incidence of mental health problem than lower SEP, why lower SEP had a higher incidence of physical illnesses than higher SEP, and why distress/ burnout had a high percentage on temporary incapacity for work,and asthma and arthrosis knee/ hip had high percentages on permanent incapacity for work. All these need further elaborated and in depth discussions.

Response 5: Thank you for this comment. We added that these differences in type of ODs reflect the prevalence of risk factors in the lower and higher SEP. ‘These findings are in line with evidence that work indeed explains socioeconomic inequalities in self-rated health among workers besides lifestyle factors [Dieker 2018] while differences in incidences of the specific ODs reflect the prevalence of risk factors in the lower SEP (e.g. noise levels [Sayler 2018] and biomechanical risk factors [Driscoll 2014] and higher SEP (psychosocial risk factors [Müller 2018].’

In Discussion (page 4, line 117-123), more references can be added to support the findings and results.

Response 6: We added extra references in the discussion section to support our findings (see also response 7).

Two      sections, Limitation and Practical implication, can be added to the paper to enrich the contents.

Response 7: Obliging to the reviewer’s remarks we extended the limitations and practical implications, i.e.

‘Furthermore, the reporting of ODs may differ between SEP, e.g. OPs are more familiar with work-related musculoskeletal disorders, which may cause selection bias. Since many ODs emerge after cumulative exposures over time, the incidence of OD may depend on age, however, in our study no large differences in age between lower and higher SEP were found at the time of the assessed ODs’.

‘The data in this paper are collected from a Dutch national registry. In future research OD incidence and incapacity for work between lower and higher SEP could be enriched if SEP at individual level encompass also educational and income level.

‘The understanding whether social position influence the possibility to stay in work with an OD have considerable interest. Although health conditions due to work hampers sustainable work ability especially in lower SEP which is also recognized in a Dutch research agenda [Knowledge agenda Prevention. Dutch National Research route Healthcare Research, Prevention and Treatment. Nederlandse Federatie van Universitair Medische Centra (NFU) en ZonMw, The Netherlands 2018], higher SEP are at risk for work related stress related disorders. The frequently reported ODs among the Dutch working population revealed population attributable fractions varying between 3% and 25%, leaving considerable potential for preventive actions on work-related exposure in terms of the occurrence of ODs and their medical and productivity cost [Van der Molen et al. 2018]’

‘These findings are in line with evidence that work partly explain socioeconomic inequalities in self-rated health among workers besides lifestyle factors [Dieker 2018], while differences in incidences of the specific ODs reflect the prevalence of risk factors in the lower SEP (e.g. noise levels [Sayler 2018] and biomechanical risk factors [Driscoll 2014] and higher SEP (psychosocial risk factors [Müller 2018].’

The conclusion (page 4) seems a bit short. A more detailed summary should be made.

Response 8: We extended on a broader summary including the answers of our research questions.

Reviewer 3 Report

I would recommend having all of the table on the same page. It is distracting and confusing to break it in half. If it has to be divided, perhaps considering re-listing the column headings on page 2.

After reading the manuscript,  I found the data source to be well described and appropriate to answer the research question.  The analytical model/method was appropriate to answer the research question.  The findings have face validity. I have no suggestions about how the research method could be improved.  I also find the paper to be well written.

Author Response

Response to reviewer 3

I would recommend having all of the table on the same page. It is distracting and confusing to break it in half. If it has to be divided, perhaps considering re-listing the column headings on page 2.

Response 1: We places the whole table on a separate place

After reading the manuscript,  I found the data source to be well described and appropriate to answer the research question.  The analytical model/method was appropriate to answer the research question.  The findings have face validity. I have no suggestions about how the research method could be improved.  I also find the paper to be well written

Response 2: Thank you for your compliment and time taken to review our manuscript.